# Growth cone advance requires EB1 as revealed by genomic replacement with a light-sensitive variant

**Alessandro Dema[1], Rabab Charafeddine[1], Shima Rahgozar[1], Jeffrey van Haren[2], Torsten Wittmann[1]***

[1]Department of Cell and Tissue Biology, University of California, San Francisco, San Francisco, United States; [2]Department of Cell Biology, Erasmus MC, Rotterdam, Netherlands

**Abstract** A challenge in analyzing dynamic intracellular cell biological processes is the dearth of methodologies that are sufficiently fast and specific to perturb intracellular protein activities. We previously developed a light-sensitive variant of the microtubule plus end-tracking protein EB1 by inserting a blue light-controlled protein dimerization module between functional domains. Here, we describe an advanced method to replace endogenous EB1 with this light-sensitive variant in a single genome editing step, thereby enabling this approach in human induced pluripotent stem cells (hiPSCs) and hiPSC-derived neurons. We demonstrate that acute and local optogenetic EB1 inactivation in developing cortical neurons induces microtubule depolymerization in the growth cone periphery and subsequent neurite retraction. In addition, advancing growth cones are repelled from areas of blue light exposure. These phenotypes were independent of the neuronal EB1 homolog EB3, revealing a direct dynamic role of EB1-mediated microtubule plus end interactions in neuron morphogenesis and neurite guidance.

***For correspondence:**
Torsten.Wittmann@ucsf.edu

**Competing interest:** The authors declare that no competing interests exist.

## Editor's evaluation

In their manuscript, Dema et al. showcase a valuable tool to study the role of the microtubule end-binding protein, EB1. This important study replaces endogenous EB1 with a light-sensitive variant, which they use to locally inactivate EB1 in human iPSC-derived neurons. They find that EB1 inactivation induces microtubule depolymerization in the growth cone and neurite retraction. The data is of high quality, the evidence supporting the conclusions is solid, and the findings of this work will be of interest to cell biologists and neurobiologists, while the methods utilized will have even broader general interest.

## Introduction

Microtubules (MTs) are essential components of the cytoskeleton in both mature neurons, in which polarized MT tracks support axonal transport, as well as during nervous system development (*Atkins et al., 2023*; *Kapitein and Hoogenraad, 2015*; *van de Willige et al., 2016*). During neuronal differentiation, the expression of many microtubule-associated proteins (MAPs) is upregulated, and numerous neurodevelopmental diseases are being linked to genetic alterations in MAPs, MT motors, and tubulin itself (*Franker and Hoogenraad, 2013*; *Maillard et al., 2023*). Developing neurites must elongate over long distances integrating extrinsic cues to achieve correct nervous system topology. Neurite growth and pathfinding are driven by adhesion and F-actin dynamics in the growth cone, the advancing terminal structure of growing neurites. Analogous to the role of MTs in migrating cells,

dynamic MTs enter the F-actin rich growth cone periphery and these so-called 'pioneer' MTs participate in growth cone advance and guidance downstream of extracellular cues (*Liu and Dwyer, 2014*; *Vitriol and Zheng, 2012*).

Proteins that associate with growing MT plus ends (+TIPs), such as spectraplakins (MACF1), the adenomatous polyposis coli protein APC, CLASPs, and neuron navigator proteins (Nav1), play critical roles in neuron morphogenesis (*Cammarata et al., 2016*; *Coles and Bradke, 2015*). For example, Nav1 mediates interactions between dynamic MT plus ends and F-actin filaments (*Sánchez-Huertas et al., 2020*) and CLASPs are involved in cell-matrix adhesion site remodeling (*Stehbens et al., 2014*). These +TIPs bind to growing MT plus ends through small adaptor proteins of the end-binding (EB) family. In humans, EBs are encoded by three MAPRE genes, and although EB1 is the best characterized +TIP adaptor, it remains unclear to what extent EB functions overlap. For example, EB3, which is upregulated in neurons and other differentiated cell types, coordinates MT and F-actin dynamics in developing neurons independent of EB1 (*Gordon-Weeks, 2017*).

The role of EB1-dependent+TIPs in neuron morphogenesis is inferred mostly from genetic knockout phenotypes in model systems from different species. It has not been tested how +TIP association with growing MT ends participates in controlling growth cone dynamics in real time in developing human neurons in part due to an absence of technology to acutely inhibit these interactions. Here, we utilize our recent optogenetic tool to disrupt EB1 function and thus +TIP MT-binding in growth cones of hiPSC-derived developing cortical neurons. In addition to demonstrating a novel strategy to generate photosensitive variants of multidomain proteins in a single genome editing step, we show that EB1 is necessary to stabilize MTs in the growth cone periphery and to maintain growth cone advance, and that this function of EB1 cannot be compensated by EB3.

## Results and discussion
### One-step genome editing to generate photosensitive protein variants

We recently reported an optogenetic system to inactivate end-binding protein 1 (EB1) with high spatial and temporal accuracy in living cells by inserting a light-sensitive LOV2/Zdk1 dimerization module between the N-terminal +TIP and C-terminal MT-binding domains of EB1 (*Figure 1A*). We named this photo-inactivated construct π-EB1 and demonstrated effects on MT dynamics and function in interphase and mitotic human cancer cells (*Dema et al., 2022b*; *van Haren et al., 2018*). However, our initial method of generating π-EB1 cell lines required sequential genetic knockouts and re-expression of photosensitive EB1 variants. To enable and demonstrate the utility of this LOV2/Zdk1-mediated multidomain splitting strategy to photo-inactivate proteins expressed at endogenous levels in more complex cell systems such as an hiPSC model of neuronal morphogenesis, we devised a CRISPR/Cas9-mediated genome editing strategy to directly insert the LOV2/Zdk1 dimerization module into the EB1 gene and thus convert endogenous EB1 into the photosensitive π-EB1 variant in one genome editing step.

Evaluating different designs of what we named π-elements in H1299 human cancer cells, we found that homozygous integration of a π-element in which an internal EF1α promoter drives expression of the C-terminal π-EB1 half while the N-terminal half remains under the control of the endogenous promoter (*Figure 1B*) resulted in balanced expression of both π-EB1 parts and in homozygous edited clones yielded π-EB1 protein levels that were similar to EB1 in control cells (*Figure 1—figure supplement 1*). In contrast, an internal ribosome entry site (IRES) resulted in very poor expression of the C-terminal half. Of note, self-cleaving 2 A peptides also did not work because the C-terminus of the LOV2 domain cannot be modified without greatly inhibiting Zdk1 binding (*Wang et al., 2016*).

We next tested π-element integration in i[3]N cells, an hiPSC line that expresses Ngn2 under the control of a doxycycline-induced promoter to allow inducible differentiation into cortical glutamatergic i[3]Neurons (*Wang et al., 2017*). Homozygous π-element integration into both copies of exon 5 of the endogenous MAPRE1 (EB1) gene was validated by genomic PCR, sequencing, and immunoblot (*Figure 1C and D*). Because the N-terminal π-EB1 half is expressed from the endogenous EB1 promoter, compared with EB1 in control cells, EB1N-LOV2 expression levels were similar and only mildly reduced in both i[3]N hiPSCs and i[3]Neurons (*Figure 1D and E*). In contrast, expression of Zdk1-EB1C that is driven by the EF1α promoter was markedly increased in i[3]N hiPSCs but was lower in i[3]Neurons (*Figure 1D*). This indicates a change in the activity balance of these two promoters

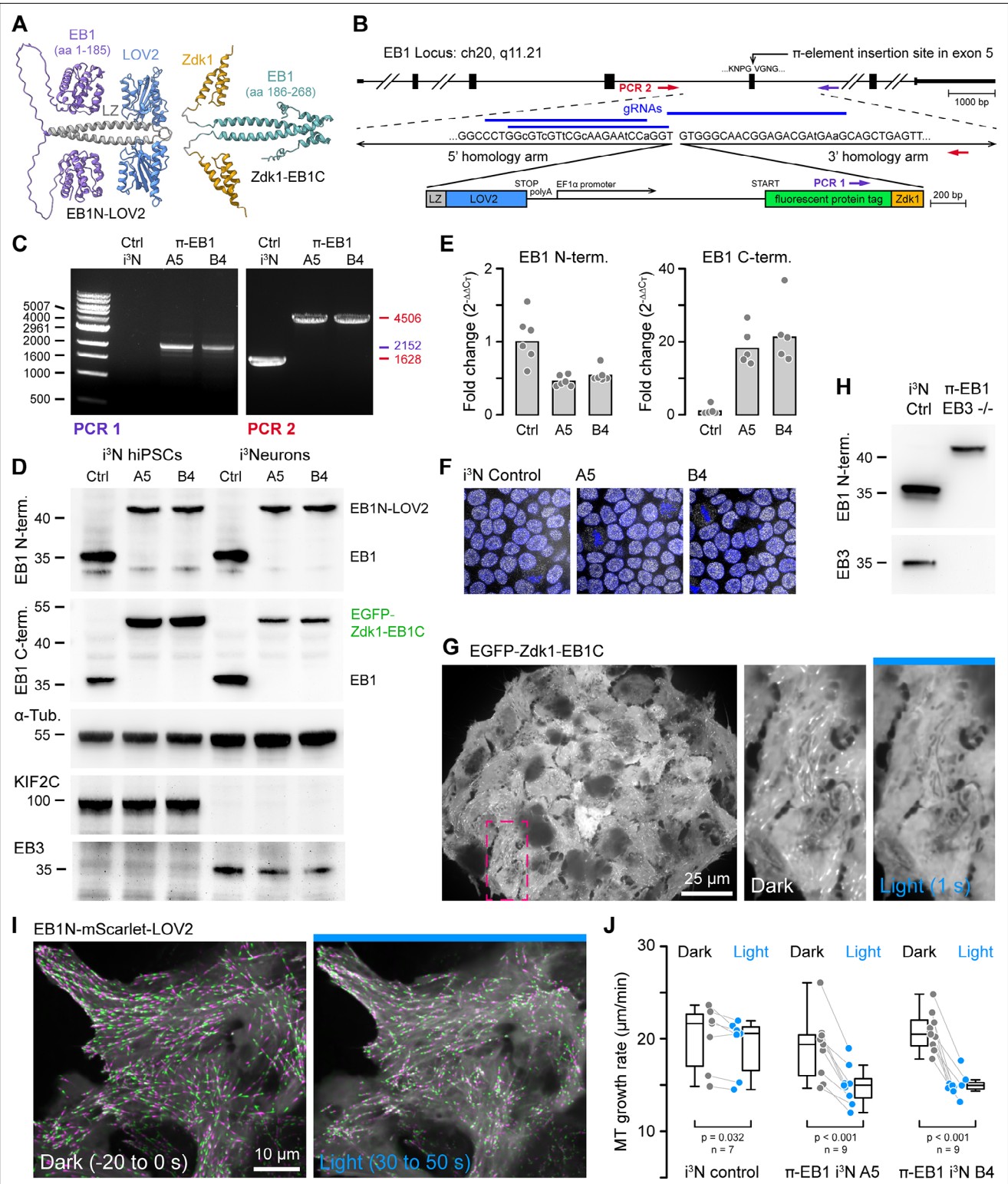

**Figure 1.** One-step genome editing to replace EB1 with a photo-sensitive variant. (**A**) AlphaFold2 model of the π-EB1 tetramer (*Mirdita et al., 2022*). Note that AlphaFold2 does not correctly predict relative domain positions and did not capture the LOV2/Zdk1 interaction correctly although a structure of the LOV2/Zdk1 dimer has previously been determined (*Wang et al., 2016*). (**B**) Overview of the one-step CRISPR/Cas9-mediated insertion of a π-element construct containing the photosensitive LOV2/Zdk1 module, a fluorescent protein marker, and an internal EF1α promoter. Arrows indicate the location of PCR primers. Lowercase letters indicate mutations introduced to make the homology-directed repair (HDR) template resistant to Cas9 cleavage. (**C**) Genomic PCR to validate π-element integration into the endogenous EB1 locus with primers as indicated in (**B**). The two clones shown

*Figure 1 continued on next page*

*Figure 1 continued*

are homozygous as there is no short product in PCR2, which corresponds to the non-edited EB1 locus. (**D**) Immunoblots with antibodies as indicated of control and π-EB1 i³N clones before and after 2 days of neuron differentiation showed replacement of EB1 by the photosensitive π-EB1 variant and expected +TIP expression level changes associated with neuron differentiation. (**E**) RT-qPCR analysis of the expression levels of the π-element N- and C-terminal halves relative to EB1 expression in wild-type (Ctrl) i³N hiPSCs. Shown are the mean and data points from individual qPCR reactions. (**F**) Comparison of nuclear Oct4 staining (white) as a pluripotency marker in control and π-EB1 i³N hiPSC colonies. Nuclei are identified with DAPI (blue). (**G**) Image of a π-EB1 i³N hiPSCs colony with magnified images on the right showing dissociation of EGFP-Zdk1-EB1C from growing MT ends in blue light. (**H**) Immunoblot of control and π-EB1 and EB3-/- i³Neurons showing expression of π-EB1 and deletion of both EB1 and EB3. (**I**) π-EB1 i³N hiPSCs transiently expressing a mScarlet-tagged EB1N MT-binding domain before and during blue light exposure. Maximum intensity projections in alternating green and magenta over 20 s at 3 s intervals illustrate attenuation of MT growth during blue light exposure. (**J**) Quantification of the median MT growth rate per cell before and during blue light exposure in control and π-EB1 i³N hiPSCs. Gray lines connect data points from the same cell. Statistical analysis by paired t-test for each i³N hiPSC line.

The online version of this article includes the following video, source data, and figure supplement(s) for figure 1:

**Figure supplement 1.** Evaluation of π-element insertion and design in H1299 lung cancer cells.

**Figure supplement 1—source data 1.** Original DNA gel and immunoblot images.

**Figure supplement 2.** Gene expression changes associated with neuron differentiation.

**Figure supplement 3.** Validation of EB3 knockout in π-EB1 i³N cells.

**Figure supplement 3—source data 1.** Original DNA gel images.

**Figure 1—video 1.** Time-lapse of genome-edited π-EB1 i³N hiPSCs transiently expressing EB1N-mScarlet-LOV2 to visualize microtubule (MT) growth dynamics before and during blue light exposure.

https://elifesciences.org/articles/84143/figures#fig1video1

**Figure 1—video 2.** Long-term phase contrast time-lapse of π-EB1 EB3-/- i³Neurons in the absence of blue light exposure showing normal neurite and growth cone dynamics.

https://elifesciences.org/articles/84143/figures#fig1video2

**Source data 1.** Original DNA gel and immunoblot images.

during i³N neuronal differentiation and suggests that the π-element design could be improved by also using the native promotor to drive expression of the C-terminal half. However, we did not try this, because proliferation, stem cell properties (*Figure 1F*), and most importantly differentiation into i³Neurons appeared normal as indicated by expected changes in expression levels of marker proteins such as EB3, DCX, and KIF2C that increase or decrease, respectively, during neuronal differentiation (*Figure 1D*, *Figure 1—figure supplement 2A* and B) (*Blair et al., 2017*). Because EB1 and EB3 are very similar and may be at least partially functionally redundant, we also removed EB3 expression by introducing a frameshift and premature stop codon near the start of the EB3 open reading frame in π-EB1 i³N hiPSCs (*Figure 1H*; *Figure 1—figure supplement 3*) to be able to ask how EB1 contributes to neuron morphogenesis and compare the relative contributions of EB1 and EB3.

To visualize π-EB1 photodissociation in genome-edited i³N cells, we chose to tag the π-EB1 C-terminal part with EGFP so that longer wavelength imaging channels remain available. Clonal π-EB1 i³N hiPSC colonies homogenously expressed EGFP-Zdk1-EB1C, and as expected, EGFP-Zdk1-EB1C rapidly dissociated from growing MT ends during blue light exposure in both i³N stem cells (*Figure 1G*) and differentiating i³Neurons (*Figure 1—figure supplement 2D*). In addition, and similar to what we previously observed in interphase human cancer cells (*van Haren et al., 2018*), π-EB1 photoinactivation significantly attenuated MT growth in i³N hiPSCs transiently expressing EB1N-mScarlet-LOV2, which remains on MT ends and enables MT growth rate measurements before and during blue light exposure (*Figure 1I and J*, *Figure 1—video 1*).

## EB1 is required to stabilize growth cone microtubule growth

Upon Ngn2-induced differentiation, π-EB1 EB3-/- i³Neurons sprouted neurites normally (*Figure 1—figure supplement 3D*, *Figure 1—video 2*), and as expected, the EGFP-tagged C-terminal half of π-EB1 associated with growing MT ends in neurites with the majority of growing MT ends localized to growth cones. Of note, because MT growth is slower in neurons than in proliferating cells (*Stepanova et al., 2003*), EGFP-Zdk1-EB1C comets were less elongated in i³Neurons and appeared more punctate compared with i³N hiPSCs. Nevertheless, EGFP-Zdk1-EB1C quickly and reversibly dissociated from MT ends upon blue light exposure in π-EB1 EB3-/- i³Neurons (*Figure 2A*, *Figure 2—video 1*).

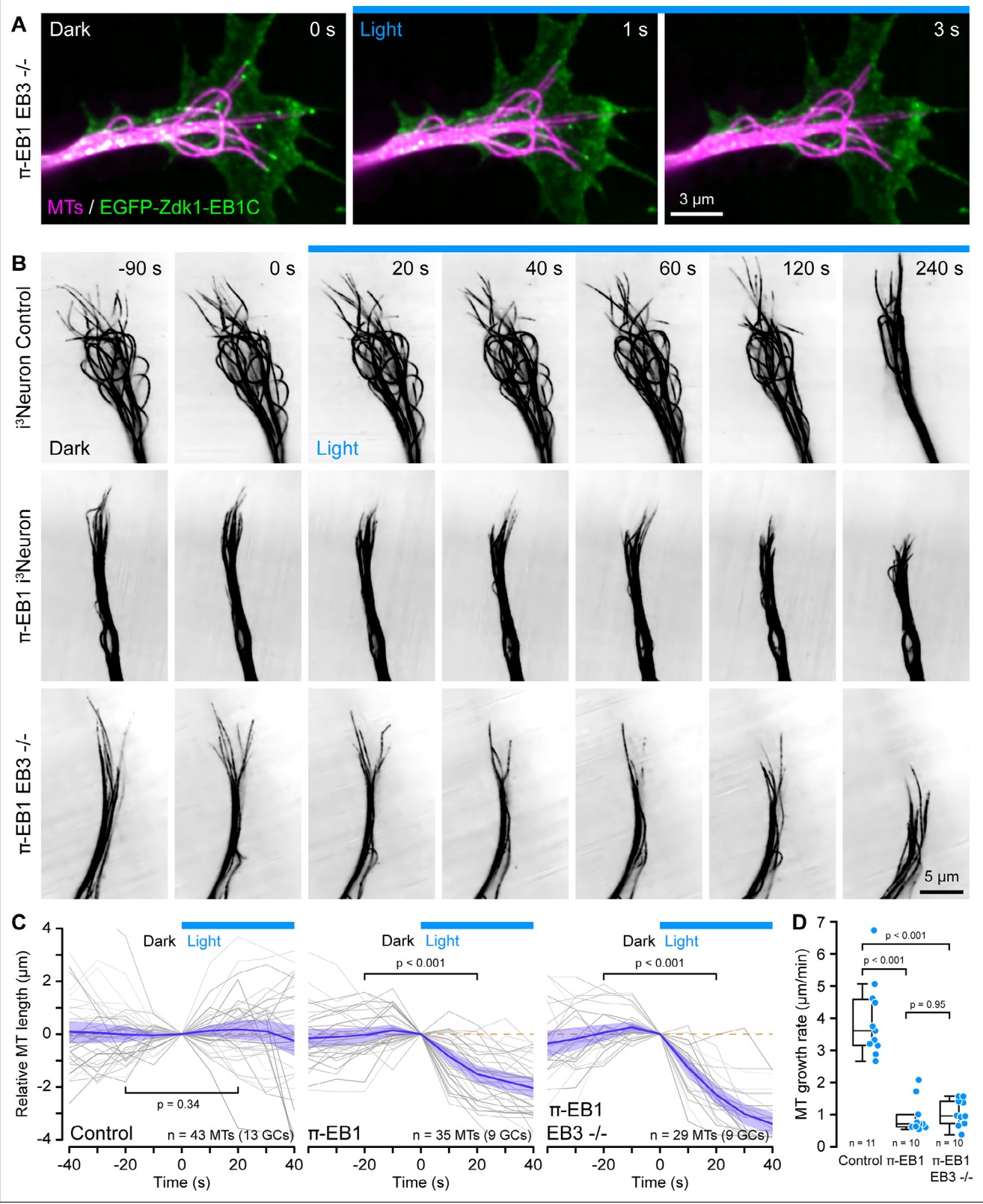

**Figure 2.** π-EB1 photoinactivation destabilizes MTs in i³Neuron growth cones. (**A**) Growth cone of a π-EB1 EB3-/- i³Neuron in which MTs were labeled with the far-red cabazitaxel derivative 4–610 CP-CTX. EGFP-Zdk1-EB1C dissociates from growing MT ends within seconds of blue light exposure. (**B**) MTs labeled with SPY555-tubulin in growth cones of i³Neurons with the π-EB1 genotype indicated on the left. Note that microtubules (MTs) continue to dynamically extend into the growth cone periphery in control i³Neurons but frequently depolymerize upon blue light exposure in both π-EB1 and π-EB1

*Figure 2 continued on next page*

*Figure 2 continued*

EB3-/- i³Neuron growth cones. Images are shown in inverted contrast for better visibility. (**C**) Quantification of the length change of 3–4 MTs per growth cone of MT ends that were clearly visible before and during blue light exposure. Gray lines are individual MTs. Blue line is the average of all MTs and the shaded area indicates the 95% confidence interval. The orange dashed line indicates no change. Statistical analysis by paired t-test at 20 s before and during blue light exposure. (**D**) Quantification of the growth cone MT growth rate after 60 s of blue light exposure by tracking SPY555-tubulin-labeled MT ends. Data points represent the average of >100 frame-to-frame growth rate measurements from multiple MTs per growth cone. Statistical analysis by ANOVA and Tukey-Kramer HSD. To better show individual growth cone MTs, the gamma of the tubulin channels was adjusted non-linearly.

The online version of this article includes the following video for figure 2:

**Figure 2—video 1.** Time-lapse of EGFP-tagged π-EB1 C-terminal half in π-EB1 EB3-/- i³Neurons demonstrating repeatability of π-EB1 photodissociation in differentiating neurons.

https://elifesciences.org/articles/84143/figures#fig2video1

**Figure 2—video 2.** Time-lapse of MT dynamics in SPY555-tubulin labeled control and π-EB1 EB3-/- i³Neuron growth cones before and during blue light exposure.

https://elifesciences.org/articles/84143/figures#fig2video2

Using a fluorescently labeled EB1 MT-binding domain, we previously reported in interphase cells and here in proliferating i³N hiPSCs that π-EB1 photoinactivation attenuated MT growth. Because we were unable to reliably transfect differentiating i³Neurons, we instead used new fluorogenic taxane derivatives as live MT labels (*Bucevičius et al., 2020*; *Lukinavičius et al., 2014*) at very low concentrations to analyze length changes of growth cone MTs that were visible both before and during blue light exposure of the entire growth cone. In control i³Neuron growth cones, MTs dynamically extended from the central domain into the growth cone periphery and the average length of these MTs was not affected by blue light. In contrast, in π-EB1 i³Neurons MTs in the growth cone periphery frequently shortened in response to blue light exposure (*Figure 2B and C*; *Figure 2—video 2*). While π-EB1 photoinactivation-induced MT shortening was most pronounced in π-EB1 EB3-/- i³Neurons, MTs also shortened in π-EB1 i³Neurons that still expressed EB3 and in both cases the relative MT length after 20 s of blue light was significantly reduced compared with control i³Neurons (p<0.001 by ANOVA and Tukey-Kramer HSD).

Although taxane-based probes label growing MT ends more dimly than the rest of the MT possibly due to slow redistribution kinetics on MTs in cells (*Ettinger et al., 2016*), we were able to measure growth cone MT polymerization rates during blue light exposure by manual tracking MT length changes. Even though MT growth rates in neurons are lower compared with other cells (*Stepanova et al., 2003*), it is important to note that absolute MT growth rates in this experiment are further underestimated because of the reduced temporal resolution compared with EB1 tracking (*Gierke et al., 2010*). Nevertheless, compared with control i³Neurons, after 1 min of blue light exposure, growth cone MT growth rates in both π-EB1 lines were reduced at least fourfold (*Figure 2D*).

Thus, taken together these data show that EB1-mediated MT end interactions are required to stabilize and sustain net growth cone MT growth. In addition, our results indicate that endogenous EB3 cannot compensate for the acute loss of EB1 function in i³Neuron growth cones. Because we previously observed that π-EB1 photoinactivation can partially displace EB3 from growing MT ends (*van Haren et al., 2018*), we cannot completely exclude a dominant negative effect of π-EB1 photoinactivation on EB3. However, because the EB1N-LOV2 expression level in i³Neurons is below that of EB1 in control cells, EB1N-LOV2 likely does not interfere with EB3 binding to MTs by direct competition for binding sites. Our findings are however consistent with not fully understood functional differences between EB1 and EB3 in developing neurons that relate both to their binding sites on MT ends as well as differences in interaction partners. Specifically, EB3 trails behind EB1 on growing MT ends and only EB3 binds to the F-actin regulator drebrin possibly coordinating F-actin and MT dynamics independently of directly controlling MT growth (*Poobalasingam et al., 2022*; *Roth et al., 2018*). Alternatively, EB3 may also be more important during later stages of neuron development when EB3 is more highly expressed (*Jaworski et al., 2009*; *Leterrier et al., 2011*).

## Acute EB1 inactivation does not immediately change F-actin dynamics

Because 'pioneer' MTs that enter the growth cone periphery participate in growth cone guidance (*Buck and Zheng, 2002*), and MT and F-actin dynamics are coupled both mechanically and biochemically through Rho GTPase signaling, we next asked how π-EB1 photoinactivation affected F-actin

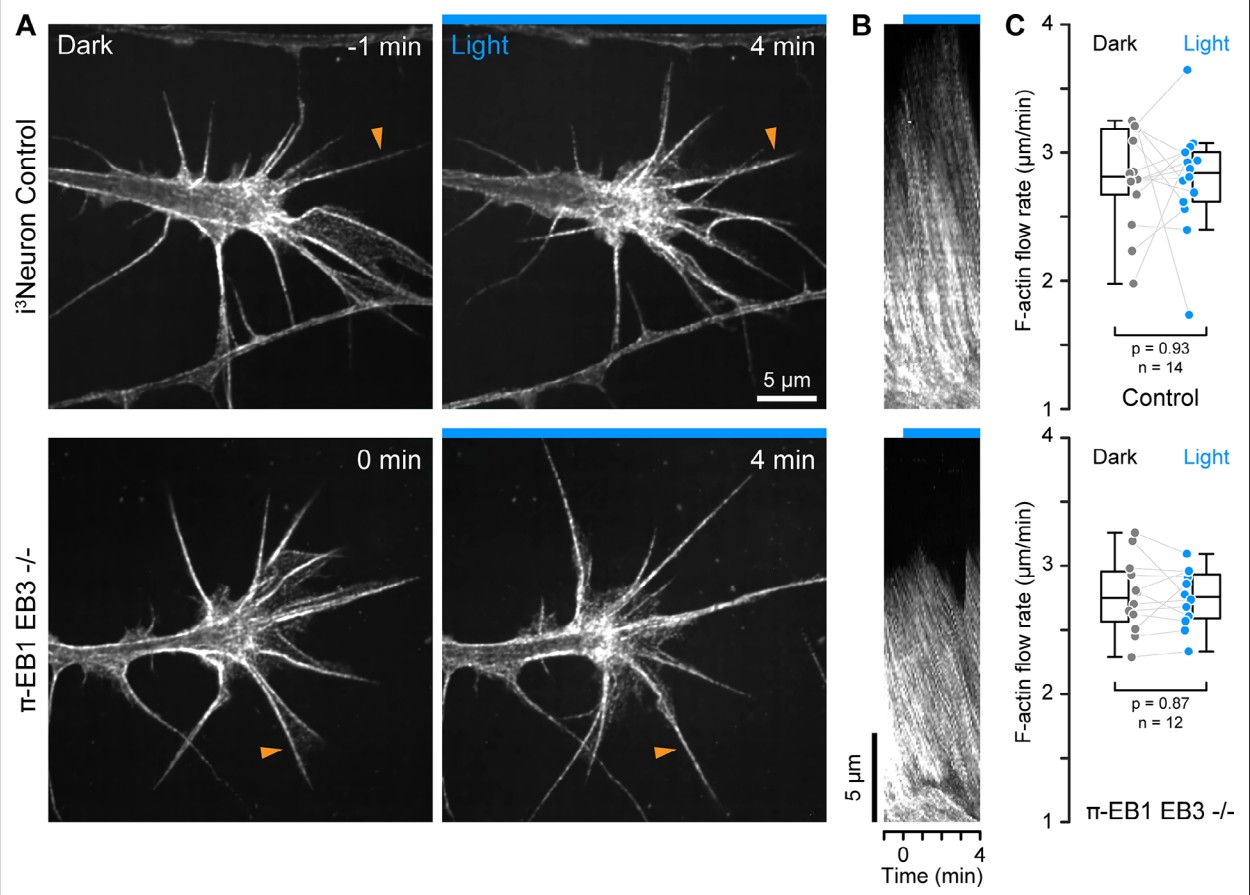

**Figure 3.** F-actin dynamics in π-EB1 neuron growth cones. (**A**) Growth cones of control and π-EB1 EB3-/- i³Neurons labeled with SPY650-FastAct before and during blue light exposure. Apparent relocalization of F-actin to the middle of the growth cone is observed in both conditions and is likely related to the photobleaching of the probe. (**B**) Kymographs along the filopodia indicated by orange arrowheads in A illustrating F-actin retrograde flow. (**C**) Quantification of the F-actin retrograde flow rate before and during blue light exposure. Each data point represents the average of at least three flow measurements per growth cone. Gray lines connect data points from the same growth cone. Statistical analysis by paired t-test. In addition, ANOVA of all four groups showed no significant difference between control and π-EB1 EB3-/- i³Neurons (p>0.97 for all pairwise comparisons with Tukey-Kramer HSD).

The online version of this article includes the following video for figure 3:

**Figure 3—video 1.** Time-lapse of a growth cone of a π-EB1 EB3-/- i³Neuron labeled with SPY650-FastAct before and during blue light exposure indicating that F-actin retrograde flow is insensitive to π-EB1 photodissociation at this time scale.

https://elifesciences.org/articles/84143/figures#fig3video1

dynamics. MT growth is thought to activate Rac1 which drives leading-edge actin polymerization, while MT shortening activates RhoA that increases actomyosin contractility through the release of regulatory MT-bound factors (*Garcin and Straube, 2019*; *Wittmann and Waterman-Storer, 2001*). To investigate growth cone F-actin dynamics in response to π-EB1 photodissociation-mediated growth cone MT shortening, we used SPY650-FastAct, a fluorogenic jasplakinolide derivative that binds strongly to F-actin filaments and at very low concentrations forms fluorescent speckles that serve as reporters of F-actin flow dynamics (*Figure 3A*; *Danuser and Waterman-Storer, 2006*). On kymographs perpendicular to the growth cone edge (*Figure 3B*), we measured the same F-actin retrograde flow rate in control (2.8+/-0.4 µm/min; mean +/-standard deviation) and π-EB1 EB3-/- i³Neurons (2.8+/-0.3 µm/min) in the dark, similar to previous measurements in growth cones from other vertebrate species (*Flynn et al., 2012*; *Geraldo and Gordon-Weeks, 2009*; *Gomez and Letourneau, 2014*). Although growth cone morphology was highly dynamic (*Figure 3—video 1*), the growth cone F-actin retrograde flow rate remained remarkably constant during blue light exposure (*Figure 3C*), and we also did not observe consistent changes in overall F-actin dynamics. Because retrograde F-actin flow is directly

related to the rate of leading-edge actin polymerization, this indicates that EB1-mediated MT growth or +TIP interactions do not immediately influence growth cone actin polymerization dynamics.

## EB1 is required for growth cone advance

To test how π-EB1 photoinactivation affected neurite dynamics, we next tracked growth cone position over longer periods of time. Within a 15 min observation window, >90% of π-EB1 neurites visibly retracted when the entire growth cone was exposed to blue light (*Figure 4A and B*; *Figure 4—video 1*), regardless of whether these cells expressed EB3 or not, again indicating that EB3 is not able to compensate for the acute loss of EB1 function in early neurite development. In contrast and as expected, control i³Neurons were insensitive to blue light exposure with only ~40% of neurites shortening. Similarly, neurites from π-EB1 i³Neurons were repelled by blue light barriers placed in front of the growth cone while control i³Neurons were unaffected (*Figure 4C*), and in long term, time-lapse experiments π-EB1 i³Neurons were unable to cross a blue light barrier sometimes after repeated growth attempts over several hours (*Figure 4D*; *Figure 4—video 2*).

Custom-engineered neuron networks with defined connectivity between individual neurons are a potential key to better understand nervous system information processing (*Aebersold et al., 2016*), and optogenetic guidance for developing neurites could be a useful tool to build such neuron networks from the bottom up. We, therefore, tested if more precise blue light exposure to small regions inside growth cones targeting only a few MTs could be used to control the direction of growth cone advance (*Figure 5A*). However, this experiment was technically very challenging. Although π-EB1 photo-dissociation remained sharply localized to small blue light-exposed regions (*Figure 5—figure supplement 1*), it was difficult to correctly place these regions in dynamic, moving growth cones. Hence, most π-EB1 EB3-/- neurites (29 out of 58) still retracted even with localized blue light exposure, but ~38% (22 growth cones, six of which also eventually retracted) turned away from the blue light-exposed region (*Figure 5*). In contrast, control i³Neurons did not respond and showed no turning bias relative to localized blue light exposure.

In summary, we present a novel approach to replace the central MT regulator EB1 with a photo-sensitive variant by inserting a light-sensitive protein interaction module encoded by a short genetic cassette – that we named π-element – into an inter-domain linker by CRISPR/Cas9-mediated genome editing. It is important to note that the π-element only needs to contain ORFs encoding LOV2 and Zdk1 and the required regulatory elements, while the flanking homology arms direct it to the correct genomic locus. The rest of the protein of interest remains encoded by the endogenous gene. Here, we also include a fluorescent protein tag to monitor π-EB1 photodissociation and a leucine zipper coiled-coil to retain dimerization of the π-EB1 N-terminal half during blue light exposure. While we demonstrate the utility of this approach for the MAPRE1 gene in hiPSCs, we believe that a similar strategy could be adapted to many other multidomain proteins supplementing the optogenetic toolbox to investigate localized protein functions in real-time (*Wittmann et al., 2020*), but it should be noted that photodissociation kinetics may be different with monomeric proteins. The LOV2/Zdk1 module interacts in the dark, which is opposite to all other optogenetic dimerization modules and therefore, enables an unperturbed dark state and acute functional knockout of a given protein activity through blue light exposure. Thus, replacing an endogenous protein with a photosensitive variant in a single genome editing step as presented here has important advantages by allowing modification of proteins for which genetic knockouts might be lethal and further enabling optogenetics in complex cell systems that are not amenable to transient genetic manipulations.

This approach allowed us to directly demonstrate that EB1 in developing human cortical neurons is required for sustained growth cone MT growth and growth cone advance. Although we do not completely dissect the molecular mechanism underlying the π-EB1 photodissociation-mediated neurite retraction response, unchanged F-actin polymerization dynamics during blue light indicate that retraction likely results from a loss of growth cone adhesion or an increase in neurite actomyosin contractility consistent with MT-shortening induced RhoA activation (*Joo and Olson, 2021*). An increase in neurite contractility that does not remain localized to the region of blue light exposure may also explain our difficulty in using π-EB1 photodissociation to control growth cone guidance more precisely. Thus, although other optogenetic tools that provide attractive stimuli could be a promising avenue to control growth cone guidance (*Harris et al., 2020*), inhibitory stimuli that induce MT depolymerization may be too difficult to control to be practically useful in synthetic biology approaches

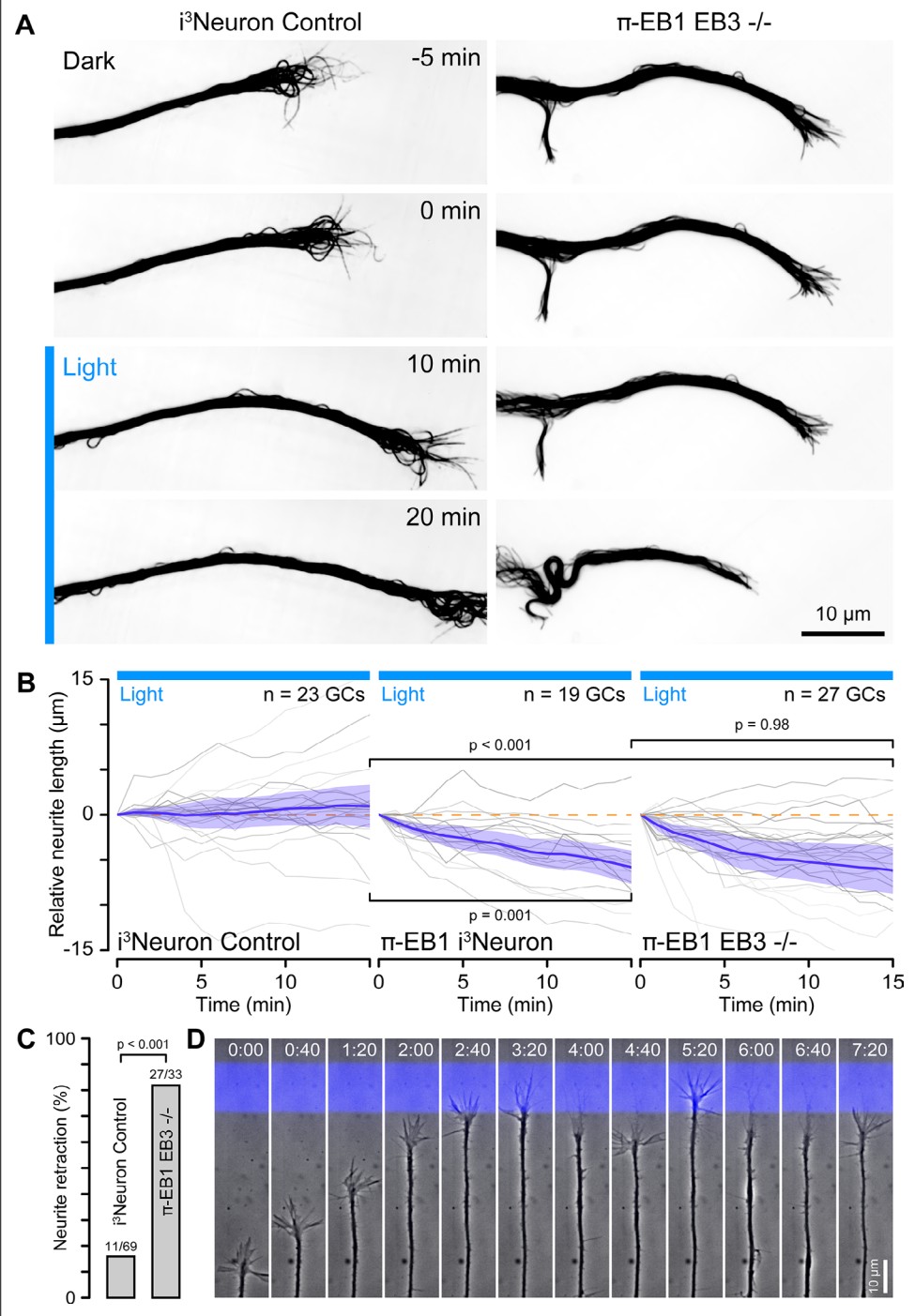

**Figure 4.** π-EB1 photoinactivation blocks growth cone advance. (**A**) Control and π-EB1 EB3-/- i³Neuron neurites in which MTs were labeled with 4–610 CP-CTX before and during blue light exposure illustrating the retraction of the π-EB1 neurite in blue light while the control neurite continues to advance. In this experiment, the entire growth cone and adjacent neurite were exposed to blue light. (**B**) Quantification of the neurite length change before and during blue light exposure. Gray lines indicate individual neurites. Blue line is the average of all neurites, and the shaded area indicates the 95% confidence interval. The orange dashed line indicates no change. Statistical analysis by ANOVA and Tukey-Kramer HSD at 15 min of blue light exposure. (**C**) Quantification of the retraction response of control and π-EB1 EB3-/- i³Neuron growth cones that encounter a blue light barrier. Statistical analysis by Fisher's exact test. (**D**) Long-term phase contrast time-lapse sequence of a π-EB1 neurite advancing upward on a 10 μm wide stripe of laminin illustrating growth cone retraction every time the growth cone attempts to cross the virtual blue light barrier. Elapsed time is indicated in hours:minutes.

*Figure 4 continued on next page*

*Figure 4 continued*

The online version of this article includes the following video for figure 4:

**Figure 4—video 1.** Time-lapse of neurite dynamics in control and π-EB1 EB3-/- i³Neurons in which MTs were labeled with 4–610 CP-CTX before and during blue light exposure.

https://elifesciences.org/articles/84143/figures#fig4video1

**Figure 4—video 2.** Long-term phase contrast time-lapse of a π-EB1 neurite advancing along laminin path toward a blue light exposed region.

https://elifesciences.org/articles/84143/figures#fig4video2

building defined neuron networks. Nevertheless, it will be interesting to see how π-EB1 in i³Neurons or other types of neurons can be used to analyze how MTs and +TIPs contribute dynamically to other aspects of neuron morphogenesis such as branching, dendritic spine dynamics or synaptic plasticity (*Dent et al., 2004*; *Jaworski et al., 2009*).

# Materials and methods

**Key resources table**

| Reagent type (species) or resource | Designation | Source or reference | Identifiers | Additional information |
|---|---|---|---|---|
| Cell line (*Homo sapiens*) | i³N induced pluripotent stem cells | *Wang et al., 2017* | | Doxycycline-induced differentiation into cortical i³Neurons |
| Cell line (*Homo sapiens*) | NCI-H1299 | ATCC | CRL-5803 | |
| Antibody | anti-EB1 (N-terminal epitope, mouse monoclonal) | Thermo Fisher Scientific | Cat# 41–2100, RRID:AB_2533500 | 1:1000 (WB) |
| Antibody | anti-EB1 (C-terminal epitope, mouse monoclonal) | BD Biosciences | Cat# 610534, RRID:AB_397891 | 1:1000 (WB) |
| Antibody | anti-EB3 (rat monoclonal) | Absea | Cat# KT36 | 1:1000 (WB) |
| Antibody | anti-KIF2C (mouse monoclonal) | Santa Cruz Biotechnology | Cat# sc-81305, RRID:AB_2132051 | 1:1000 (WB) |
| Antibody | anti-Oct-3/4 (mouse monoclonal) | Santa Cruz Biotechnology | Cat# sc-5279, RRID:AB_628051 | 1:500 (IF) |
| Recombinant DNA reagent | EB1N-LZ-LOV2 (plasmid) | *van Haren et al., 2018* | Addgene plasmid 107614 | |
| Recombinant DNA reagent | MSCV-PIG | Scott Lowe, unpublished | Addgene plasmid 18751 | IRES plasmid |
| Recombinant DNA reagent | pmCherry-Zdk1-EB1C | *van Haren et al., 2018* | Addgene plasmid 107695 | |
| Recombinant DNA reagent | pSpCas9(BB)–2A-GFP | *Ran et al., 2013* | Addgene plasmid 48138 | Cas9 plasmid |
| Sequence-based reagent | EB1 Exon 1–2 | Integrated DNA Technologies | RT-qPCR primers, Hs.PT.58.1854993 | |
| Sequence-based reagent | EB1 Exon 6–7 | Integrated DNA Technologies | RT-qPCR primers, Hs.PT.58.24290642 | |
| Sequence-based reagent | EB3 | Integrated DNA Technologies | RT-qPCR primers, Hs.PT.58.20604386 | |
| Sequence-based reagent | DCX | Integrated DNA Technologies | RT-qPCR primers, Hs.PT.58.118505 | |

*Continued on next page*

*Continued*

| Reagent type (species) or resource | Designation | Source or reference | Identifiers | Additional information |
|---|---|---|---|---|
| Chemical compound, drug | SPY555-tubulin | Spyrochrome / Cytoskeleton Inc. | CY-SC203 | 1:2000 |
| Chemical compound, drug | SPY650-FastAct | Spyrochrome / Cytoskeleton Inc. | CY-SC505 | 1:3000 |
| Chemical compound, drug | 4–610 CP-CTX | *Bucevičius et al., 2020* | | 5 nM |

## Molecular cloning and genome editing

### Construction of knock-in π-EB1 cassettes

Primer sequences for all cloning, gRNAs, and genomic PCR are given in *Supplementary file 1*.

1. pEB1N-LZ-LOV2-IRES-mCherry-Zdk1-EB1C was cloned as follows: EB1N-LZ-LOV2 (*Wittmann and van Haren, 2018*) (Addgene plasmid 107614) and EMCV IRES (from MSCV-PIG; Addgene plasmid 18751) were amplified by PCR, connected by overlap extension PCR, and ligated into the NheI site of pmCherry-Zdk1-EB1C (Addgene plasmid 107695).

2. pUC19-π-EB1_IRES-mCherry (LZ-LOV2-IRES-mCherry-Zdk1) containing the homology-directed repair (HDR) template for targeting the IRES containing a version of the π-element into exon 5 of MAPRE1/EB1 was cloned as follows: >1.5 kb 5'and 3' homology arms were amplified from genomic DNA isolated from one 6 cm dish of immortalized human retinal pigment epithelial cells (hTERT-RPE1 cells, ATCC) using a Purelink Genomic DNA Mini Kit (ThermoFisher). The π-element was PCR amplified from plasmid #1. Homology arms and π-element sequences were cloned into the EcoRI site of pUC19 by Gibson assembly in a single step. Several silent mutations were introduced to ensure that the HDR template could not be targeted by the gRNAs.

3. pUC19-sv40-blastR-EF1α-mCherry-Zdk1-EB1C was generated by Gibson assembly of PCR amplified SV40promoter-BlasticidinR, SV40polyA, EF1alpha promoter (from a PiggyBac vector) and mCherry-Zdk1-EB1C-polyA (from pmCherry-Zdk1-EB1C) into HinDIII and EcoRI sites of pUC19.

4. pEB1N-LZ-LOV2-polyA-EF1alpha-mCherry-Zdk1-EB1C was generated by Gibson assembly of PCR amplified polyA-EF1alpha-mCherry-Zdk1-EB1C from plasmid #3 into the BamHI site of pEB1N-LZ-LOV2.

5. pUC19-π-EB1_EF1alpha-mCherry (LZ-LOV2-polyA-EF1α-mCherry-Zdk1) containing the HDR template for targeting the π-element to exon 5 of MAPRE1/EB1 was cloned by Gibson assembly of PCR amplified homology arms and π-element into the EcoRI site of pUC19 using the same primers as for the IRES plasmid (#2).

6. For generating π-EB1 knock-in hiPSCs an EGFP-tagged variant of the π-element was used. pUC19-π-EB1_EF1α-EGFP was generated by replacing mCherry in plasmid #5 with EGFP by Gibson assembly of PCR-amplified EGFP into the NcoI sites.

### Guide RNAs

7. pSpCas9(BB)–2A-GFP was a gift from Feng Zhang (Addgene plasmid 48138). pSpCas9(BB)–2A-mCherry was constructed by replacing GFP with mCherry. T2A-mCherry was PCR amplified and inserted into the EcoRI sites by Gibson assembly.

8. Three gRNA sequences that target <25 bp from the integration site in exon 5 of MAPRE1/EB1 were cloned into the BbsI sites of pSpCas9(BB)–2A-GFP (*Wittmann and van Haren, 2018*) to knock in the π-element into the EB1 gene.

9. Similarly, three gRNA sequences targeting exon 1 of MAPRE3 were cloned into the BbsI sites of pSpCas9(BB)–2A-GFP and pSpCas9(BB)–2A-mCherry to generate EB3 knockout lines.

### Other DNA constructs

10. EB1N-mScarlet-I-LZ-LOV2 and EB1N-mApple-LZ-LOV2 were cloned by inserting PCR amplified mScarlet-I/mApple and LZ-LOV2 into the XhoI and BamHI sites of EB1-GFP-LOV2 by Gibson

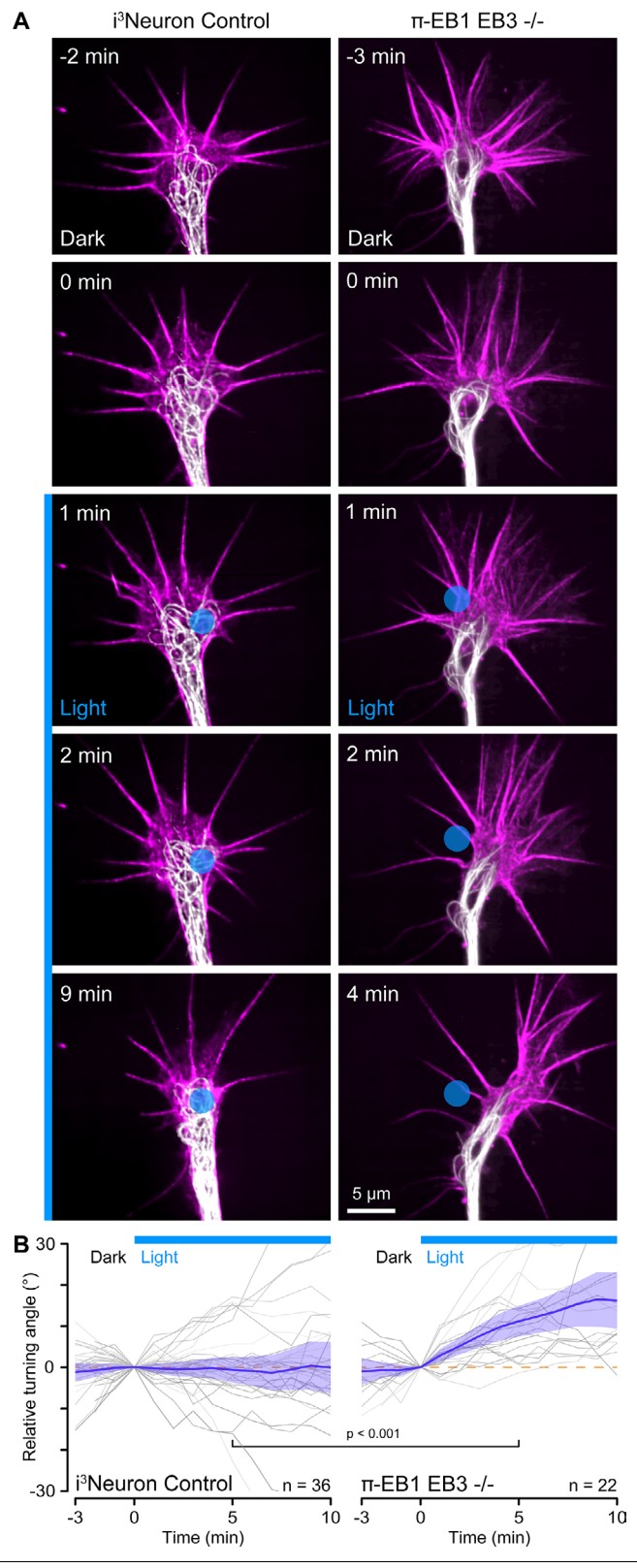

**Figure 5.** Growth cone turning in response to local π-EB1 photoinactivation. (**A**) Time-lapse of control and π-EB1 EB3-/- i³Neuron growth cones labeled with SPY555-tubulin (white) and SPY650-FastAct (magenta). The gamma of the tubulin channel was adjusted to 0.6 to better visualize growth cone microtubules (MTs). The blue circle indicates the light-exposed area. (**B**) Quantification of the relative turning angle in response to local blue light

*Figure 5 continued on next page*

*Figure 5 continued*

exposure. Gray lines are individual growth cones. Blue line is the average of all growth cones measurements, and the shaded area indicates the 95% confidence interval. The orange dashed line indicates the 0° angle. Statistical analysis by unpaired t-test at 5 min of local blue light exposure.

The online version of this article includes the following figure supplement(s) for figure 5:

**Figure supplement 1.** Accuracy of local π-EB1 photodissociation.

---

assembly. While the original EB1N-mCherry-LZ-LOV2 fusion construct did not perform well in iPSC-derived neurons (and showed striking aggregation), the mScarlet and mApple fusions showed the expected localization pattern at MT plus-ends. The two resulting PCR products were inserted into the XhoI/BamHI sites of plasmid EB1N-GFP-LOV2. mScarlet-I was PCR amplified using a mRuby2 forward oligo, which resulted in a slightly altered mScarlet-I N-terminus (MVSKGEEL instead of MVSKGEAV).

## Cell lines

H1299 human small lung cancer cells were authenticated by STR profiling. i³N hiPSCs are derived from the WTC11 human induced pluripotent stem cell line (*Wang et al., 2017*). No STR profile is available. Cells tested negative for mycoplasma.

## i³Neuron cell culture and differentiation

i³N hiPSCs were cultured essentially as described (*Wang et al., 2017*). In brief, i³N cells were cultured on dishes coated with Matrigel (08774552, Corning) in mTeSR1 medium (85850, STEMCELL Technologies) changed every 2 days for 3–5 days, until 70–80% confluent, then gently detached with Accutase (07920, STEMCELL Technologies) and reseeded at $2–5×10^4$ cells/cm² in mTeSR1 medium containing 10 µM Y-27632 (72304, STEMCELL Technologies) ROCK inhibitor (removed after 48 hr and exchanged for a fresh medium without ROCK inhibitor). i³N cells were transfected with Lipofectamine Stem (STEM00015, Invitrogen) 24 hr after seeding according to the manufacturer's protocol.

For pre-differentiation, $2–3×10^6$ i³N hiPSCs were seeded on a Matrigel-coated 3.5 cm dish (high crowding helps the early steps of the differentiation process) in knockout DMEM/F12 (12660012, Life Technologies) supplemented with 1 x NEAA (11140050, Life Technologies), 1x N2 supplement (17502048, Life Technologies), 2 µg/ml doxycycline, 10 ng/ml recombinant neurotrophin-3 (NT-3, 50712487, Biolegend), 10 ng/ml recombinant brain-derived neurotrophic factor (BDNF, NC1043629, Fisher Scientific), 0.3 µg/ml murine laminin (CB40232, Corning) and 10 µM Y-28632 ROCK inhibitor (only for the first day, then withdrawn). The medium was changed every 24 hr for 3 days of culture.

For imaging-directed differentiation, 20 mm glass-bottom dishes (P35G-1.5–20 C, Mattek) were surface-activated for 1–2 min in a Harrick Expanded Plasma Cleaner (*Dema et al., 2022a*) set on 'High' and coated with 50 µg/ml poly-D-lysine (PDL, A-003-E, Sigma) in PBS for 20 min at 37 °C, washed extensively in PBS, then coated with 1.5 µg/ml murine laminin in PBS for 20 min at 37 °C. After a further PBS wash, $2×10^4$ to $10^5$ pre-differentiated i³N cells from the Accutase-dissociated cell pellet were seeded in Maturation Medium (MM), consisting of 50% Neurobasal (12349015, Life Technologies) and 50% DMEM/F12 (21041025, Life Technologies) supplemented with 1 x NEAA, 0.5 x GlutaMAX (35050061, Life Technologies), 1x N2 supplement, 1 x B27 supplement (17504044, Life Technologies), 2 µg/ml doxycycline, 10 ng/ml NT-3, 10 ng/ml BDNF and 0.3 µg/ml murine laminin. Laminin micropatterns on glass-bottom dishes were generated as described (*Dema et al., 2022a*). For immunoblots or qPCR, the pre-differentiated cells were seeded at $10^5$ cm⁻² on PDL and laminin-coated tissue culture plastic dishes. Cell lysate preparation and immunoblotting for EB1 and EB3 were performed as described (*Wittmann and van Haren, 2018*).

## Antibodies

Mouse monoclonal anti-EB1 (N-terminal epitope): Thermo Fisher Scientific Cat# 41–2100, RRID:AB_2533500; Mouse monoclonal anti-EB1 (C-terminal epitope): BD Biosciences Cat# 610534, RRID:AB_397891; Rat monoclonal anti-EB3: Absea Cat# KT36, no RRID available; Mouse monoclonal anti-KIF2C: Santa Cruz Biotechnology Cat# sc-81305, RRID:AB_2132051; Mouse monoclonal anti-Oct-3/4: Santa Cruz Biotechnology Cat# sc-5279, RRID:AB_628051.

## Genome editing

In general, cells were co-transfected with two or three gRNA Cas9 plasmids targeting MAPRE1/EB1 exon 5 and a pUC19 HDR template described above. H1299 cells were transfected using Lipofectamine 2000 in a 10 cm dish at ~75% confluency and 2–3 days later mCherry-positive cells (indicating π-element integration) were FACS sorted into 96-well plates for expansion and further analysis.

For generating i3N π-element lines, both the π-element and gRNA plasmids carried EGFP markers. Thus, cells were FACS sorted 6 days after transfection when we expected that most cells had lost transient expression from the gRNA plasmids. EGFP-positive i3N cells were then plated at low density (~3000 cells) in 6 cm Matrigel-coated dishes and cultured in mTeSR with ROCK inhibitor as described above. After 7–10 days, clonal colonies typically had grown enough to be seen by the naked eye. Up to 96 colonies from each dish were manually picked and expanded.

Initial screening for positive clones in both H1299 and i3N cells was by spinning disk confocal microscopy because clones in which fluorescent signal was present on MT plus ends in the absence of blue light must have correctly integrated the π-element such that both halves are expressed as the N-terminal half (π-EB1N) expressed from the endogenous promoter is needed to target EGFP-tagged π-EB1C to MT plus ends. Together with wild-type controls, such clones were further analyzed by genomic PCR with primer sets that bind inside and outside the π-element sequence (*Supplementary file 1*). Genomic DNA was isolated using a Purelink Genomic DNA Mini Kit (ThermoFisher). Expected PCR products for EGFP EF1α π-element: primer pair 11 a/11b, WT: 189 bp, π-element integration: 3058 bp; 11 c/11b, WT: 1628 bp; π-element integration: 4506 bp; 11e/11 f, WT: no product, π-element integration: 2152 bp.

For the MAPRE3/EB3 knockout in π-element i3N cells, gRNAs were cloned into a mCherry version of the Cas9 plasmid, and transfected cells FACS-sorted for mCherry expression, plated at low density, and clones expanded as described above. Initial positive clones identified by band shift in genomic PCR (see *Supplementary file 1* for primer sequences) were further analyzed by Sanger sequencing of the PCR amplicon and analyzed by ICE (Synthego Performance Analysis, ICE Analysis. 2019. v3.0. Synthego). Pre-mixed prime-time qPCR primers were from IDT.

## Live microscopy and π-EB1 photoinactivation

i3Neuron imaging experiments were performed 1–3 days after replating on laminin-coated glass-bottom dishes. At later times i3Neurons form intricate networks and it becomes increasingly difficult to locate individual growth cones. SPY555-tubulin and SPY650-FastAct (Spyrochrome) were added to i3Neurons at a 1:2000 and 1:3000 dilution, respectively, and 4–610 CP-CTX was added at 5 nM for at least 30 min before imaging, and cells were discarded after a maximum of 3 hr.

Live microscopy was performed either with a Yokogawa CSU-X1 spinning disk confocal essentially as described (*Stehbens et al., 2012*) or, for most i3Neuron microscopy, with a CFI Apochromat TIRF 60 X NA 1.49 objective (Nikon) on a Yokogawa CSU-W1/SoRa spinning disk confocal system, and images acquired with an ORCA Fusion BT sCMOS camera (Hamamatsu). For high-resolution imaging of dim signal, SoRa mode was combined with 2x2 camera binning resulting in an image pixel size of 54 nm. This system was equipped with a Polygon 1000 pattern illuminator (Mightex) through an auxiliary filter turret and LAPP illuminator (Nikon). Integrated control of imaging and photoinactivation was through NIS Elements v5.3 software (Nikon), in combination with an external pulse generator to trigger the 470 nm photoinactivation LED (Spectra X light engine, Lumencor) for 10 ms pulses at 2 Hz at 5–10% LED power. π-EB1 photoinactivation was confirmed by dissociation of the C-terminal half from growing MT ends. The Polygon was calibrated with a mirror slide before every imaging session, and local light exposure was further verified by imaging the back reflection from the specimen coverslip.

## Image analysis and statistics

MT growth rate analysis in i3N hiPSCs was as described (*van Haren et al., 2018*). MT ends in SPY555-tubulin labeled i3Neurons as well as neurite length changes were tracked manually using the 'Segmented Line' tool of FiJi. Kymographs to measure retrograde F-actin flow were generated as described (*Stehbens et al., 2014*). Details of statistical analysis including the type of test, p-values, and numbers of biological replicates are provided within the relevant figures and figure legends. Statistical analysis was done in MATLAB (Mathworks, Inc) and graphs were produced in MATLAB and

in Excel (Microsoft). In all figures, box plots show the median, first, and third quartile, with whiskers extending to observations within 1.5 times the interquartile range.

## Acknowledgements

We thank Li Gan for the i³N hiPSCs, and Gražvydas Lukinavičius and Jonas Bucevičius for the generous gift of fluorogenic cabazitaxel MT probes. We also thank Phillip Gordon-Weeks for reminding us of the differences in EB1 and EB3 binding sites near MT ends. This work was supported by National Institutes of Health grants R21 CA224194, R01 NS107480, S10 RR026758 and S10 OD028611 to TW.

## Additional information

### Funding

| Funder | Grant reference number | Author |
| --- | --- | --- |
| National Cancer Institute | R21 CA224194 | Torsten Wittmann |
| National Institute of Neurological Disorders and Stroke | R01 NS107480 | Torsten Wittmann |
| National Institutes of Health | S10 RR026758 | Torsten Wittmann |
| National Institutes of Health | S10 OD028611 | Torsten Wittmann |

The funders had no role in study design, data collection and interpretation, or the decision to submit the work for publication.

### Author contributions

Alessandro Dema, Jeffrey van Haren, Conceptualization, Formal analysis, Investigation, Writing - original draft, Writing - review and editing; Rabab Charafeddine, Shima Rahgozar, Investigation; Torsten Wittmann, Conceptualization, Formal analysis, Funding acquisition, Writing - original draft, Project administration, Writing - review and editing

### Author ORCIDs

Alessandro Dema ⓘ http://orcid.org/0000-0003-0976-9396
Jeffrey van Haren ⓘ http://orcid.org/0000-0002-3160-3547
Torsten Wittmann ⓘ http://orcid.org/0000-0001-9134-691X

### Decision letter and Author response

Decision letter https://doi.org/10.7554/eLife.84143.sa1
Author response https://doi.org/10.7554/eLife.84143.sa2

## Additional files

### Supplementary files

• Supplementary file 1. Excel spreadsheet detailing all primer and guide RNA sequences used for cloning, CRISPR/Cas9 genome editing, and genomic PCR.
• MDAR checklist

### Data availability

Raw data have been deposited to Dryad: doi:https://doi.org/10.7272/Q6CF9NC5.

The following dataset was generated:

| Author(s) | Year | Dataset title | Dataset URL | Database and Identifier |
|---|---|---|---|---|
| Wittmann T, Dema A, Charafeddine R, Rahgozar S, van Haren J | 2023 | Data for: Growth cone advance requires EB1 as revealed by genomic replacement with a light-sensitive variant | https://doi.org/10.7272/Q6CF9NC5 | Dryad Digital Repository, 10.7272/Q6CF9NC5 |

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
