## [Editor Report]

In their manuscript, Dema et al. showcase a valuable tool to study the role of the microtubule end-binding protein, EB1. This important study replaces endogenous EB1 with a light-sensitive variant, which they use to locally inactivate EB1 in human iPSC-derived neurons. They find that EB1 inactivation induces microtubule depolymerization in the growth cone and neurite retraction. The data is of high quality, the evidence supporting the conclusions is solid, and the findings of this work will be of interest to cell biologists and neurobiologists, while the methods utilized will have even broader general interest.

---

## [Decision Letter]

**Decision letter after peer review:**

Thank you for submitting your article "Growth cone advance requires EB1 as revealed by genomic replacement with a light-sensitive variant" for consideration by *eLife*. Your article has been reviewed by 3 peer reviewers, and the evaluation has been overseen by a Reviewing Editor and Anna Akhmanova as the Senior Editor. The following individuals involved in the review of your submission have agreed to reveal their identity: Laura Anne Lowery (Reviewer #1); Feng-Quan Zhou (Reviewer #3).

Essential revisions:

1) Every reviewer agreed that this new methodology is compelling, but that the functional analysis required additional controls to strengthen the study. Specifically, the authors need to perform controls for Figure 3, Figure 4, and Figure 5. Please refer to the comments by Reviewers #1 and #3 for specific suggestions. In addition, please be sure to provide quantification of the control conditions.

2) All of the reviewers highlighted the need for additional significance tests. Please be sure to perform statistical analyses for every experiment and include in the legend the n, the statistical test used, and the p-value.

*Reviewer #1 (Recommendations for the authors):*

The manuscript would benefit from editing for grammar to improve readability. For example, on p.3, a comma is missing after "neuron navigator proteins (Nav1)" and before "play." Another example – the last sentence of p. 4 is a run-on sentence that spans 7 lines and has incorrect grammar. These are just two examples.

p.3 – +TIP is not spelled out during its first use.

p.5 – referral to Figure 1F,G in text should be Figure 1G,H instead.

Figure 2/Figure legend p. 22 – In the legend, "MTs labelled with SPY555-tubulin in growth cones of i3Neurons with the π-EB1 genotype indicated on the right." This should be "on the left."

p.6/Figure 2 – Text reads "In contrast, in π-EB1 i3Neurons MTs rapidly retracted from the peripheral domain in response to blue light exposure (Figure 2B, C; Video 4). While this MT retraction was most pronounced in π-EB1 EB3-/- i3Neurons, MTs also shortened in π-EB1 i3Neurons that still expressed EB3, and in both cases this π-EB1 photoinactivation-induced MT shortening was significantly different from control i3Neurons."

Unfortunately, the chosen representative neuron for the π-EB1 EB3-/- condition in Figure 2 does not appear to have significantly shorter MTs in the light compared to in the dark. In fact, to this reviewer, the MTs in the final right panel (the final light condition) appear to be actually longer than the third panel (the final dark condition). Thus, this image does not appear to be representative nor fit the data depicted in the graph of Figure 2C. One potential issue/reason for this is that the MTs in the EB3-/- growth cones are already compromised, but this is not mentioned.

p.7/Figure 5 – Testing whether precise blue light exposure to small regions inside growth cones targeting only a few MTs could be used to control the direction of growth cone advance – They reported in the text that 42 out of 58 cells still retracted, with only localized exposure. Then, they reported that "analyzing GC displacement of neurites that did not immediately retract revealed a turning response away from the blue light exposed area (Figure 5B)." However, in Figure 5B, they quantify the results of 22 growth cones. The numbers here do not add up. 42 cells that retract plus 22 cells that do not retract equals 64, not 58. Thus, are there some cells that retracted but not immediately and were counted in the turning quantification? Also, are there some cells that did not retract at all and were not counted? This should be clarified.

*Reviewer #2 (Recommendations for the authors):*

While this is a well-conducted study, the publication of which I would support, there are some details I would like to see addressed before approval for publication:

Fig1S: These data are not sufficient to support the claim that expression of the N-terminal half of the π-EB1 is comparable to control cells. The WBs have to be quantified. Additionally, qPCR should be used to compare expression levels.

Figure 3C: What statistical test has been used here? If it is a t-test, a p>0.05 does not mean the values are the same, it just means that these data are not sufficient to detect a difference. In other words, you cannot reject the hypothesis that they are from the same distribution, which does not mean that they are from the same distribution. Additionally, n=8 is fairly low for this type of analysis. The strength of this experiment would be increased by increasing the n.

Figure 4A: Where was the neurite illuminated? It looks like the largest change happens somewhat down the shaft. Was the whole neurite illuminated? Please indicate in the figure as well as in the text.

Figure 5: Please also show this experiment in the control neurons.

*Reviewer #3 (Recommendations for the authors):*

The authors should at least quantify the presented results with statistical analysis and add the necessary control experiments mentioned in the public review.

A better explanation with new experiments or discussion about the results regarding EB3 is also necessary.

It will greatly improve the study if the new approach used in the study could discover some novel function of EB1 or related cellular mechanism underlying its function.

---

## [Author Response]

Essential revisions:1) Every reviewer agreed that this new methodology is compelling, but that the functional analysis required additional controls to strengthen the study. Specifically, the authors need to perform controls for Figure 3, Figure 4, and Figure 5. Please refer to the comments by Reviewers #1 and #3 for specific suggestions. In addition, please be sure to provide quantification of the control conditions.

We have now included additional control experiments. Figure 3: F-actin retrograde flow in control growth cones and additional data points for the π-EB1 growth cones; Figure 4: Quantification of control and π-EB1 growth cone response encountering a blue light barrier; Figure 5: Control growth cones are now included for the turning experiment. Additional detail is given below in the responses to specific criticism.

2) All of the reviewers highlighted the need for additional significance tests. Please be sure to perform statistical analyses for every experiment and include in the legend the n, the statistical test used, and the p-value.

We have now included additional control experiments. Figure 3: F-actin retrograde flow in control growth cones and additional data points for the π-EB1 growth cones; Figure 4: Quantification of control and π-EB1 growth cone response encountering a blue light barrier; Figure 5: Control growth cones are now included for the turning experiment. Additional detail is given below in the responses to specific criticism.

Reviewer #1 (Recommendations for the authors):The manuscript would benefit from editing for grammar to improve readability. For example, on p.3, a comma is missing after "neuron navigator proteins (Nav1)" and before "play." Another example – the last sentence of p. 4 is a run-on sentence that spans 7 lines and has incorrect grammar. These are just two examples.p.3 – +TIP is not spelled out during its first use.p.5 – referral to Figure 1F,G in text should be Figure 1G,H instead.Figure 2/Figure legend p. 22 – In the legend, "MTs labelled with SPY555-tubulin in growth cones of i3Neurons with the π-EB1 genotype indicated on the right." This should be "on the left."

Thank you for pointing out these typos. We believe we fixed them all (while probably introducing new ones).

p.6/Figure 2 – Text reads "In contrast, in π-EB1 i3Neurons MTs rapidly retracted from the peripheral domain in response to blue light exposure (Figure 2B, C; Video 4). While this MT retraction was most pronounced in π-EB1 EB3-/- i3Neurons, MTs also shortened in π-EB1 i3Neurons that still expressed EB3, and in both cases this π-EB1 photoinactivation-induced MT shortening was significantly different from control i3Neurons."Unfortunately, the chosen representative neuron for the π-EB1 EB3-/- condition in Figure 2 does not appear to have significantly shorter MTs in the light compared to in the dark. In fact, to this reviewer, the MTs in the final right panel (the final light condition) appear to be actually longer than the third panel (the final dark condition). Thus, this image does not appear to be representative nor fit the data depicted in the graph of Figure 2C. One potential issue/reason for this is that the MTs in the EB3-/- growth cones are already compromised, but this is not mentioned.

We do not entirely agree with this assessment and think that the difference in growth cone MT response to blue light was evident in the previous version of Figure 2. (i.e. fewer growth cone MT ends during blue light in the π-EB1 lines). Nevertheless, we redesigned Figure 2 to better illustrate blue light-induced growth cone MT shortening in the π-EB1 lines. In addition to different example growth cones, we also rotated the images vertically and added additional panels during blue light to better show MT shortening. In our quantification, the difference with or without EB3 is very small and not fully consistent, and although we cannot completely exclude a slight effect of EB3, we do not think that EB3 substantially contributes to growth cone MT growth. While this was initially surprising to us, this is consistent with EB3 not binding to the same site as EB1, but rather trailing the EB1 comet. We expanded the discussion of functional differences between EB1 and EB3 and would like to acknowledge Phillip Gordon-Weeks, who pointed this out in a comment on the bioRxiv preprint.

p.7/Figure 5 – Testing whether precise blue light exposure to small regions inside growth cones targeting only a few MTs could be used to control the direction of growth cone advance – They reported in the text that 42 out of 58 cells still retracted, with only localized exposure. Then, they reported that "analyzing GC displacement of neurites that did not immediately retract revealed a turning response away from the blue light exposed area (Figure 5B)." However, in Figure 5B, they quantify the results of 22 growth cones. The numbers here do not add up. 42 cells that retract plus 22 cells that do not retract equals 64, not 58. Thus, are there some cells that retracted but not immediately and were counted in the turning quantification? Also, are there some cells that did not retract at all and were not counted? This should be clarified.

The reviewer is correct, the reason why the numbers did not add up is exactly because there were a few cells that first turned and then retracted. We clarified this in the text.

Reviewer #2 (Recommendations for the authors):While this is a well-conducted study, the publication of which I would support, there are some details I would like to see addressed before approval for publication:Fig1S: These data are not sufficient to support the claim that expression of the N-terminal half of the π-EB1 is comparable to control cells. The WBs have to be quantified. Additionally, qPCR should be used to compare expression levels.

The main point of Figure 1 – supplement 1 was to show in a simple non-neuronal cell line that an IRES did not work well to drive sufficient expression of the C-terminal π-element half and thus provide a rationale why we did not attempt this in hiPSCs. We did not mean to indicate that π-EB1 expression levels are identical to EB1 in control cells, merely that they are similar enough (as opposed to the IRES construct in which there is essentially no mCherry-Zdk1-EB1C expression). We clarified this in the text. While we do not think that quantification of these immunoblots adds substantially to the manuscript, we did, however, include a genomic PCR analysis of the same clones as on the immunoblot showing that only clone #1 was edited correctly and homozygous, while clone #2 was likely heterozygous showing reduced mCherry-Zdk1-EB1C expression both on the immunoblot and by quantification of cell fluorescence intensity. Arguably more important is the π-element expression level in hiPSCs and derived neurons, and we have added an RT-qPCR analysis to Figure 1 quantitatively comparing π-EB1 in edited hiPSCs to EB1 expression in control cells and modified the text accordingly.

Figure 3C: What statistical test has been used here? If it is a t-test, a p>0.05 does not mean the values are the same, it just means that these data are not sufficient to detect a difference. In other words, you cannot reject the hypothesis that they are from the same distribution, which does not mean that they are from the same distribution. Additionally, n=8 is fairly low for this type of analysis. The strength of this experiment would be increased by increasing the n.

We have included both control i^3^Neurons as well as additional data points for the π-EB1 EB3 -/- growth cones. Paired t-tests before and during blue light as well as ANOVA of all four experimental groups show no statistically significant difference. The reviewer is correct of course that not being able to reject the null hypothesis that all samples belong to the same distribution does not automatically mean that there is no difference, but in our experiment, we cannot measure a difference in F-actin retrograde flow.

Figure 4A: Where was the neurite illuminated? It looks like the largest change happens somewhat down the shaft. Was the whole neurite illuminated? Please indicate in the figure as well as in the text.

In Figure 4A, the entire field-of-view was exposed to blue light. We now note this in the figure legend. Given that local blue light exposure in the growth cone also frequently resulted in neurite retraction (Figure 5), we think that retraction results from a loss of growth cone adhesion rather than a process in the neurite. However, we did not analyze this further in this work.

Figure 5: Please also show this experiment in the control neurons.

We have included control i^3^Neurons in the turning experiment.

Reviewer #3 (Recommendations for the authors):The authors should at least quantify the presented results with statistical analysis and add the necessary control experiments mentioned in the public review.

We have included controls and statistical analysis as requested.

A better explanation with new experiments or discussion about the results regarding EB3 is also necessary.

As outlined above, we have expanded our discussion on differences between EB1 and EB3.

It will greatly improve the study if the new approach used in the study could discover some novel function of EB1 or related cellular mechanism underlying its function.

Yes. We agree that there is significant potential in using this approach to further analyze neuron morphogenesis beyond growth cone dynamics. However, after years of COVID-related disruptions, we currently lack the resources in both funding and personnel to dive deeper. Yet, we believe it is important to get this new methodology published and shared with the community.